# Oxidative Stress Response Kinetics after 60 Minutes at Different Levels (10% or 15%) of Normobaric Hypoxia Exposure

**DOI:** 10.3390/ijms241210188

**Published:** 2023-06-15

**Authors:** Clément Leveque, Simona Mrakic Sposta, Sigrid Theunissen, Peter Germonpré, Kate Lambrechts, Alessandra Vezzoli, Maristella Gussoni, Morgan Levenez, Pierre Lafère, François Guerrero, Costantino Balestra

**Affiliations:** 1Environmental, Occupational, Aging (Integrative) Physiology Laboratory, Haute Ecole Bruxelles-Brabant (HE2B), 1160 Brussels, Belgium; c.leveque.research@gmail.com (C.L.); klambrechts@he2b.be (K.L.); morganlevenez@gmail.com (M.L.); plafere@he2b.be (P.L.); 2Laboratoire ORPHY, Université de Bretagne Occidentale, UFR Sciences et Techniques, 93837 Brest, France; francois.guerrero@univ-brest.fr; 3Institute of Clinical Physiology, National Research Council (CNR), 20162 Milan, Italy; simona.mrakicsposta@cnr.it (S.M.S.); alessandra.vezzoli@cnr.it (A.V.); 4DAN Europe Research Division (Roseto-Brussels), 1160 Brussels, Belgium; pgermonpre@gmail.com; 5Hyperbaric Centre, Queen Astrid Military Hospital, 1120 Brussels, Belgium; 6Institute of Chemical Sciences and Technologies “G. Natta”, National Research Council (SCITEC-CNR), 20133 Milan, Italy; maristella.gussoni@unimi.it; 7Anatomical Research and Clinical Studies, Vrije Universiteit Brussels (VUB), 1090 Brussels, Belgium; 8Motor Sciences Department, Physical Activity Teaching Unit, Université Libre de Bruxelles (ULB), 1050 Brussels, Belgium

**Keywords:** hypoxia, oxygen biology, cellular reactions, human, oxygen therapy, human performance, decompression

## Abstract

In this study, the metabolic responses of hypoxic breathing for 1 h to inspired fractions of 10% and 15% oxygen were investigated. To this end, 14 healthy nonsmoking subjects (6 females and 8 males, age: 32.2 ± 13.3 years old (mean ± SD), height: 169.1 ± 9.9 cm, and weight: 61.6 ± 16.2 kg) volunteered for the study. Blood samples were taken before, and at 30 min, 2 h, 8 h, 24 h, and 48 h after a 1 h hypoxic exposure. The level of oxidative stress was evaluated by considering reactive oxygen species (ROS), nitric oxide metabolites (NOx), lipid peroxidation, and immune-inflammation by interleukin-6 (IL-6) and neopterin, while antioxidant systems were observed in terms of the total antioxidant capacity (TAC) and urates. Hypoxia abruptly and rapidly increased ROS, while TAC showed a U-shape pattern, with a nadir between 30 min and 2 h. The regulation of ROS and NOx could be explained by the antioxidant action of uric acid and creatinine. The kinetics of ROS allowed for the stimulation of the immune system translated by an increase in neopterin, IL-6, and NOx. This study provides insights into the mechanisms through which acute hypoxia affects various bodily functions and how the body sets up the protective mechanisms to maintain redox homeostasis in response to oxidative stress.

## 1. Introduction

Acute hypoxia (AH) is an environmental condition that is regularly encountered, for example, by mine workers or telescope operators going to high altitudes, or by pilots flying at high altitudes [1]. Intermittent acute hypoxia is also a therapeutic approach that is of growing interest in scientific research because of its (often unrecognized) beneficial effects, as well as its adverse effects. Indeed, low-dose repetitive AH protocols have demonstrated numerous benefits, such as reducing hypertension [2], reducing inflammation [3], improving aerobic capacity [4], increasing bone mineral density [5], and improving memory [6,7,8,9] and cardiovascular function [10,11]. Furthermore, the positive effects of low-dose AH do not seem to be associated with detectable negative effects, such as systemic inflammation [12]. However, in order to optimize the use of AH as a therapeutic approach, it is important to maximize its benefits while avoiding any adverse effects. Hypoxia can have deleterious effects during stroke or cancer [13] and may negatively affect vascular function [14] or even cause altitude sickness in healthy mountaineers [15]. Of course, these harmful effects generally result from oxygen deprivation at lower oxygen fractions or for longer periods [16].

At the same time, it has been shown that intermittent variations in the inspired oxygen level, whether to hypoxia or hyperoxia, can also lead to multiple effects such as an increase in hemoglobin [17] or the stimulation of hypoxia-inducible factor 1-alpha, inflammatory markers such as nuclear factor kappa B and interleukin-6 (IL-6), antioxidant proteins as nuclear factor erythroid-2-related factor 2 (NRF2) and micro-RNAs [18].

Some authors have attempted to use hypoxia during training sessions to achieve an increased training benefit [19,20] by coupling high-intensity exercise sessions with simulated altitude. However, incorporating hypoxia into all high-intensity interval sessions had little effect on performance compared with normoxia training [21,22]. Other studies suggest that short-term exposure to hypoxic air during intense training can favorably remodel skeletal muscle [23,24,25]. However, there is no visible benefit to endurance performance, particularly in well-trained athletes [26].

It is known that reactive oxygen species (ROS) and the cellular redox state play a major role in modulating many signaling pathways. In moderate amounts, ROS are important for physiological processes, leading to positive cellular adaptive responses, while large amounts of ROS can damage lipids, proteins, and DNA, and lead to pathological responses [18]. Therefore, the exposure of healthy individuals or patients to severe hypoxia could potentially generate high levels of ROS and facilitate disease progression [27]. In contrast, a training program consisting of 15 to 24 sessions of intermittent exposure to severe hypoxia has gained popularity as a treatment in patients with a variety of chronic conditions [28,29,30]. Currently, the best protocol for acute exposure to severe hypoxia (in combination with normoxia or hyperoxia) to reach the best outcomes associated with a beneficial change in the redox status is not known [16,31,32]. Depending on the duration and severity of the hypoxia exposure, the effects on the different cellular functions can be positive or deleterious. It is therefore important to understand its kinetics after a single exposure before implementing intermittent hypoxia sessions for patient treatment or for training reasons, since the time between sessions is crucial to achieve optimal outcomes and different cell reactions could be targeted depending on the dose and repetitions [33,34]. The objective of this study is thus to observe the effects of oxy-inflammation over time in response to a single exposure to normobaric hypoxia at two different inspired fractions of oxygen (FiO_2_): 10% and 15%.

## 2. Results

### 2.1. ROS and NOx Rate, Antioxidant Response (TAC), and 8-Isoprostane (8-iso-PGF2α) Levels after One Hour of Oxygen Exposure at an FiO_2_ of 10% and 15%

Both levels of oxygen exposure, severe (10%) and mild (15%), elicited a significant increase in plasmatic ROS production rate, with a steeper increase and slower reduction for severe exposure (Figure 1A). Exposure to severe hypoxia produced more ROS than mild exposure and showed a pattern characterized by a significant increase, with a peak after 30 min for both 10% hypoxia (0.32 ± 0.03 µmol·min^−1^ compared with baseline; *p* = 0.0015; size effect = 0.16) and 15% hypoxia (0.27 ± 0.02 µmol·min^−1^; *p* < 0.001; size effect = 0.24). This peak plateaus for about 2 h quantity of ROS decreases slowly until 48 h. The difference between the two levels of exposure disappeared after 24 h (*p* = 0.87). The antioxidants (TAC) (Figure 1B) decreased, with a nadir at 30 min for the 10% FiO_2_ group (1.52 ± 0.22 mM; *p* = 0.0017; size effect = 0.50) and at 2 h for the 15% FiO_2_ group (1.70 ± 0.20 mM; *p* = 0.0010; effect size = 0.2496). Antioxidants decreased more rapidly after severe hypoxia than after mild hypoxia (*p* = 0.0191). Notably, 8-isoprostane (pg/mg creatinine) (Figure 1C) followed the exact same trend for both levels of oxygen exposure, with a peak at 2 h (553 ± 199 pg·mg^−1^ creatinine after 10% hypoxia; *p* = 0.0461; size effect = 0.64 vs. 489 ± 161 pg·mg^−1^ creatinine after 15% hypoxia; *p* = 0.0046; size effect = 0.35). After 48 h, the values returned to baseline for both levels of exposure (10%: 259 ± 91 pg·mg^−1^ creatinine; *p* = 0.4499 and 15%: 325 ± 103 pg·mg^−1^ creatinine; *p* > 0.9999). Nitric oxide metabolites (Figure 1D) showed a peak 2 h after acute exposure to 10% oxygen compared with baseline (516 ± 132 µM; *p* = 0.013; size effect = 0.51), while no significant difference was observed after exposure to 15% oxygen (*p* = 0.1303; size effect = 0.27).

### 2.2. Inflammatory Response (IL-6, Neopterin, Creatinine, and Uric Acid) after One Hour of Oxygen Exposure at an FiO_2_ of 10% or 15%

Interleukin 6 (IL-6) was measured in plasma samples while neopterin, creatinine, and urates were obtained from urine samples (Figure 2 and Figure 3).

IL-6 showed a significant increase compared with the baseline, until reaching its peak at 8 h for the 10% group (2.7 ± 0.57 pg/mL; *p* = 0.0203; effect size = 0.72). Contrary to severe hypoxia, mild hypoxia (15%) showed a significant increase from 8 h post-exposure (2.5 ± 0.28 pg/mL; *p* = 0.0179), with a peak at 24 h (2.7 ± 0.34 pg/mL; *p* = 0.0001), and slowly decreased until 48 h.

Neopterin showed a similar trend during the first 30 min for mild (15%) and severe (10%) hypoxia and reached its peak at 2 h for severe hypoxia (130.6 ± 25.6 µmol/mol creatinine; *p* = 0.1475; effect size = 0.52) and at 8 h for mild hypoxia (153.0 ± 61.1 µmol/mol creatinine; *p* = 0.0441; effect size = 0.13). The values returned to baseline after 48 h for the two levels of hypoxia.

Compared with the baseline, creatinine and urates in urine exhibited an opposite U-shaped response, with creatinine’s acme at 2 h post-exposure (10%: 1.11 ± 0.32 g/L; *p* = 0.0192; effect size = 0.17 vs. 15%: 1.68 ± 0.47 g/L; *p* = 0.0488; effect size = 0.10) and urates’ nadir between 30 min and 2 h for severe hypoxia (10%) (30 min: 4.22 ± 0.96 mg/dL; *p* = 0.3312; 2 h: 4.0 ± 0.78 mg/dL; 0.3427; effect size = 0.13) and between 8 h and 24 h for mild hypoxia (15%) (8 h: 4.23 ± 0.89 mg/dL; *p* = 0.0041; 24 h: 4.33 ± 1.5 mg/dL; *p* = 0.0084; size effect = 0.18) (Figure 3).

### 2.3. Discomfort Perceived—Visual Analog Scale (VAS)

No subject developed malaise, tiredness, headaches, sleepiness, or nausea upon exposure to one hour of hypoxia, neither with 10% nor with 15% exposure. However, hypoxia at 10% (equivalent to an altitude of approximately 5700 m) was felt to be less comfortable than that at 15% (roughly 2500 m altitude) (81.0 ± 10.6 % of declared discomfort vs. 12.9 ± 5.4% of declared discomfort, respectively; *p* > 0.001).

## 3. Discussion

The presented data confirm the results of previous studies by highlighting a significant increase in ROS production [35,36,37,38] during a short period of moderate or extreme hypoxia. We observed a faster and larger increase in ROS production as hypoxia increased. However, after 48 h of recovery, ROS levels returned close to the baseline. It is known that the mitochondrial complex I is inhibited by intermittent hypoxia through the activation of NOx (NADPH oxidase), thereby increasing the production of ROS [39]. In various situations, electrons escaping from enzymatic and nonenzymatic reactions initiate the production of ROS, represented by superoxide anion (O_2_^•−^), hydrogen peroxide (H_2_O_2_), and hydroxide (HO^•^) [40]. In an aqueous solution, O_2_^•−^ has a half-life of approximately 4 μsec, which gives it the ability to scatter over a distance ranging from 150 to 220 nm [41,42]. Thus, O_2_^•−^ can react at various locations beyond where it is generated and, as a result, can affect surrounding molecules and organelles more extensively than HO^•^. However, this distance is insufficient for extracellularly produced O_2_^•−^ to move within a cell. Therefore, O_2_^•−^ generated inside the cell has the ability to damage various cellular components, including DNA, organelles, and cellular membranes’ phospholipids. Our results are in agreement with this assumption since we observed an increase in 8-iso-PGF2α 2 h after the two levels of exposure, slightly delayed after the increase in ROS occurring 30 min after exposure.

It has been demonstrated that acute hypoxic conditions induce mitochondrial fission [43]. This reduction in mitochondrial fusion is further described in hypoxia–reoxygenation situations and is caused by decreased ATP production [44]. This adaptation appears to be a countermeasure to maintaining mitochondrial ROS production at a balanced level [45]. Furthermore, we observed an increase in NOx production, with a peak after 2 h and still significant after 8 h. Due to its radical nature, the reaction between superoxide and NO proceeds in a limited manner through diffusion (with a rate constant of k = 1.9 × 1010 M^−1^s^−1^) [46], which is faster than the SOD-catalyzed dismutation of superoxide. The advantage of trapping superoxide ions is to eliminate the radical electrons at an early stage, thereby halting subsequent radical chain reactions [40]. Indeed, our results suggest that the peak of NOx at 2 h occurring after the peak of ROS (30 min) indicates an early activity of NO as a ROS scavenger. This is further confirmed by the nonsignificant changes in NOx during normobaric hypoxia. This is consistent with previous findings [35], where the activity of GPX and SOD was not significantly increased after normobaric hypoxia [47]. Considering the abundant production of NO, the ability of NO to scavenge superoxide appears to be comparable to the activity of intracellular SOD and may even be more effective. However, the resulting ONOO- produced from this reaction is a powerful oxidant and is toxic to cells when produced in excess beyond the antioxidant capacity [48]. However, currently, new lines of evidence suggest a beneficial action of ONOO-. In the vascular system, for instance, moderate levels of ONOO- appear to stimulate prostaglandin synthesis and play a role in cellular signal transduction reactions [49]. Furthermore, under inflammatory conditions, the simultaneous production of NO and ONOO- suggests that the detoxification of superoxide by NO exceeds the cytotoxic action of ONOO- [50].

Although H_2_0_2_ was not directly measured, we hypothesize that the increase in NOx, in particular NO_2_, may reflect the onset of an adaptive activity by angiogenesis [51].

In this study, we observed a decrease in urinary uric acid (UA) excretion, more clearly expressed after 15% of oxygen breathing. This suggests a consumption of uric acid as an antioxidant.

In fact, an evolutionary advantage of UA has been proposed by demonstrating its strong capacity to eliminate free radicals, making it an excellent scavenger [52]. Studies have shown that UA contributes up to 60% to the elimination of free radicals in human serum [53]. Furthermore, the systemic administration of UA has been associated with an increase in plasma antioxidant capacity, both at rest and after exercise, in healthy subjects [54,55]. Interestingly, after 48 h, we observed a return to baseline levels, suggesting that UA plays an antioxidant role without shifting to its function as a pro-oxidant [56].

Moreover, we observed an increase in urinary creatinine excretion, with a zenith 2 h after exposure to 10% and 15% hypoxia. This suggests an increase in creatine catabolism. Although the antioxidant mechanism of creatine is not yet fully understood, it has been shown to contribute to cellular homeostasis, particularly as a mitochondrial protector [57].

In our experimental setting, no muscular activity was present. Therefore, we believe creatinine excretion is a marker of its antioxidant activity. In fact, it is known that, in response to acute high-altitude exposures with muscular activity, hyperventilation causes alkalosis. The kidneys compensate for this alkalosis by excreting excess bicarbonate and retaining hydrogen ions to reduce respiratory alkalosis [58]. Our results are then in agreement with the putative (not yet fully known) mechanism of action of creatine as an antioxidant. However, it has been shown to increase the activity of antioxidant enzymes and the capability to eliminate ROS and reactive nitrogen species [59,60,61]. We indeed found a globally symmetrical reaction to creatinine on ROS production, NOx, neopterin, and 8-iso-PGF2α.

Interestingly, we observed an increase in the initiation of 8-iso-PGF2α, which is a marker of lipid peroxidation by ROS [62,63], at 2 h post-exposure. Lipid peroxidation has its peak between 2 and 8 h; this relatively late reaction may be understood because of creatin action (cf. supra).

Our study also showed an immune system stimulation. Indeed, the increasing concentration of neopterin can be monitored in medicine to evaluate the level of clinical inflammation resulting from physical trauma, cardiovascular diseases, cancer, and bacterial, parasitic, and/or viral infections [64,65,66,67,68]. A rapid increase in neopterin was also observed (after 2 h) following the ROS peak without pathological reasons since the hypoxic stimulus was of short duration and no other harmful situations (trauma, cardiovascular diseases, etc.) were present. This rapid reaction is of interest, as it would counteract the inflammasome complex favoring the NRF2 pathway, as already shown following mild “oxy-inflammation” of 60 min [69]. This result is very interesting in the context of preconditioning protocols with the aim of inhibiting inflammasomes [70].

Interestingly, neopterin modulation is inversely proportional to the importance of hypoxia. Indeed, we observed a greater increase in neopterin after exposure to 15% oxygen than after exposure to 10% oxygen. We speculate that a higher hyperoxia-induced increase in ROS production could contribute to this difference, as already recently shown in moderate hyperoxia compared with hypoxia (oxy-inflammation) [14].

All the renal countermeasures described seem to have an efficient antioxidant role in these short hypoxic exposures together with significant TAC variations. This was not found during longer hypoxic exposures [71].

Acute hypoxia stimulates the production of IL-6 and the polarization of M1 macrophages. IL-6 is increased via the induction of antioxidant response element and nuclear factor kappa B (NF-κB) [72]. Furthermore, IL-6 (staying significantly high for 48 h) activates the NRF2 signaling pathway, which allows for the expression of antioxidant genes and maintains redox homeostasis [73]. Furthermore, it has recently been shown that NRF2 plays a key role in providing the preconditions for normobaric hypoxia to enhance exercise endurance [74].

## 4. Limitations

Strengths:–This study is, to our knowledge, one of the first to tackle the kinetic responses to a single short normobaric oxygen exposure at 10% and 15% of FiO_2_;–The measurements were taken until 48 h post-exposure and putatively open the avenue to new possible applications for hypoxic protocols;–The electron paramagnetic resonance (EPR) method used for ROS analysis is the actual gold standard.

Weaknesses:
–The number of subjects was limited, but the sample can be considered homogenous since all were healthy participants;–The analysis was not carried out in the nucleus of the cells but in the plasma. This could be considered a weakness for some, but it would need a thoroughly different experimental setting.

## 5. Materials and Methods

### 5.1. Experimental Protocol

After written informed consent, 48 healthy nonsmoking Caucasian subjects (32 males and 16 females) volunteered for this study. None of them had a history of previous cardiac abnormalities or were under any cardio- or vasoactive medication.

All experimental procedures were conducted in accordance with the Declaration of Helsinki [75] and approved by the Ethics Committee approval from the Bio-Ethical Committee for Research and Higher Education, Brussels (No. B200-2020-088).

After medical screening to exclude any latent morbidity, participants were prospectively randomized into six groups of 6–8 persons each. These groups were divided into hypoxia (10% and 15% of FiO_2_), normobaric hyperoxia (30% and 100% of FiO_2_ [34], and hyperbaric hyperoxia (1.4 ATA and 2.5 ATA) groups. All participants were asked to refrain from strenuous exercise for 48 h before the tests. No antioxidant nutrients, i.e., dark chocolate, red wine, or green tea, were permitted 8 h preceding and during the study. The subjects were also asked not to dive 48 h before the experiment and not to fly within 72 h before the experiment.

Fourteen participants were subjected to mild hypoxia (15%, *n* = 8) and extreme hypoxia (10%, *n* = 6) protocols. As far as age (10%: 34.0 ± 13.7 years old (mean ± SD) vs. 15%: 30.9 ± 13.8 years old; *p* > 0.05), height (10%: 168.2 ± 11.0 cm vs. 15%: 169.9 ± 9.8 cm; *p* > 0.05), weight (10%: 58.7 ± 21.3 kg vs. 15%: 63.8 ± 12.2 kg; *p* > 0.05), gender ratio, and health status are concerned, groups were comparable.

Oxygen-depleted air (oxygen partial pressure: 0.1 bar; 100 hPa, *n* = 6 and 0.15 bar; 150 hPa, *n* = 8) was administered for 1 h by means of an orofacial nonrebreather mask with a reservoir bag, care being taken to fit and tighten the mask on the subject’s face.

Hypoxic gas at the two levels (10% and 15%) was supplied using a hypoxia generator and set to reach the chosen level of oxygen (HYP 123, Hypoxico–Hypoxico Europe GmbH, Bickenbach, Germany). Both levels of exposure flow were calibrated by means of an oximeter (Solo-O_2_ Divesoft, Halkova, Czech Republic) in the mask used by the subject to ascertain that the desired oxygen level was reached [34].

Blood and urine samples were obtained before exposure (T0) and 30 min, 2 h, 8 h, 24 h, and 48 h after the end of oxygen administration (Figure 4). Previous experiments have shown that cellular responses after different oxidative exposures may take up to 24 h or even more. Therefore, we decided to take blood samples up to 48 h [14,18,69,76].

Each blood sample consisted of approximately 15 mL of venous human blood collected in lithium heparin and EDTA tubes (Vacutainer, BD Diagnostic, Becton Dickinson, Italia S.p.a., Florence, Italy). Plasma and red blood cells (RBCs) were separated via centrifugation (Eppendorf Centrifuge 5702R, Darmstadt Germany) at 1000× *g* at 4 °C for 10 min. The samples were then stored in multiple aliquots at −80 °C until assayed; analysis was performed within one month of sample collection.

Urine was collected by voluntary voiding in a sterile container and stored in multiple aliquots at −20 °C until assayed and thawed only before analysis.

### 5.2. Blood Sample Analysis

#### 5.2.1. Determination of ROS Using Electron Paramagnetic Resonance (EPR)

An electron paramagnetic resonance instrument (E-Scan—Bruker BioSpin, GmbH, Rheinstetten, Germany) X-band, with a controller temperature at 37 °C interfaced to the spectrometer, was adopted for ROS production rate, as already performed by some of the authors [77,78,79,80]. EPR measurements are highly reproducible, as previously demonstrated [81]. EPR is the only noninvasive technique suitable for the direct and quantitative measurement of ROS. In particular, the spectroscopic technique (EPRS) is used in many fields of application, including biomedicine [82]. The reliability and reproducibility of EPR data obtained using the micro-invasive EPR method adopted in this study have already been reported previously [83].

Briefly, for ROS detection, 50 µL of plasma was treated with an equal volume of CMH (1-hydroxy-3-methoxycarbonyl-2,2,5,5-tetramethylpyrrolidine), and then 50 µL of this solution was placed inside a glass EPR capillary tube in the spectrometer cavity for data acquisition. A stable radical CP (3-Carboxy-2,2,5,5-tetramethyl-1-pyrrolidinyloxy) was used as an external reference to convert the obtained ROS values into absolute quantitative values (μmol/min). All EPR spectra were generated by adopting the same protocol and obtained by using a standard software program supplied by Bruker (Billerica, MA, USA) (version 2.11, WinEPR System).

#### 5.2.2. Total Antioxidant Capacity (TAC)

The 6-hydroxy-2,5,7,8-tetramethylchroman-2-carboxylic acid (Trolox)-equivalent antioxidant capacity assay, a widely used kit-based commercial method, was used. Briefly, 10 uL of plasma was added in duplicate to 10 µL of metmyoglobin and 150 µL of the chromogen solution; then, reactions were initiated through the addition of 40 µL of H_2_O_2_, as indicated by the instructions (No. 709001, Cayman Chemical, Ann Arbor, MI, USA). Reaction mixtures were incubated at room temperature for 3 min and then read by measuring the absorbance signal at 750 nm using an Infinite M200 microplate reader spectrophotometer (Tecan, Grödig, Austria). A linear calibration curve was computed from pure Trolox-containing reactions.

### 5.3. Urine Sample Analysis

#### 5.3.1. Nitric Oxide Metabolites (NO_2_+NO_3_)

NOx (NO_2_+NO_3_) concentrations were determined in urine via a colorimetric method based on the Griess reaction, using a commercial kit (Cayman Chemical, Ann Arbor, MI, USA), as previously described [84]. Samples were spectrophotometrically read at 545 nm.

#### 5.3.2. 8-Isoprostane (8-iso-PGF2α)

Levels of 8-iso-PGF2α were measured using an immunoassay EIA kit (Cayman Chemical, Ann Arbor, MI, USA) in urine. This is a biomarker for lipid peroxidation and damage assessment. Samples and standards were spectrophotometrically read at 412 nm. Results were normalized using urine creatinine values.

#### 5.3.3. Interleukin-6

IL-6 levels were determined using an ELISA assay kit (ThermoFisher Scientific, Waltham, MA, USA), based on the double-antibody “sandwich” technique in accordance with the manufacturer’s instructions.

All the above samples and standards were read using a microplate reader spectrophotometer (Infinite M200, Tecan Group Ltd., Männedorf, Switzerland). The obtained values were assessed in duplicate, and the inter-assay coefficient of variation was in the range indicated by the manufacturer.

#### 5.3.4. Creatinine, Neopterin, and Uric Acid Concentrations

Urinary creatinine, neopterin, interleukin 6 (IL-6), and uric acid concentrations were measured via high-pressure liquid chromatography (HPLC), as previously described [18], using a Varian instrument (pump 240, autosampler ProStar 410, SpectraLab Scientific Inc., Markham, ON, Canada) coupled to a UV–VIS detector (Shimadzu SPD 10-AV, λ = 240 nm, SpectraLab Scientific Inc., for creatinine and uric acid; and JASCO FP-1520, λ_ex_ = 355 nm and at λ_em_ = 450 nm, SpectraLab Scientific Inc., for neopterin).

After the urine centrifugation of 1500× *g* at 4 °C for 5 min, analytic separation procedures were performed at 50 °C on a 5 µm Discovery C18 analytical column (250 × 4.6 mm I.D., Supelco, Sigma-Aldrich, Merck Life Science S.r.l., Milano, Italy) at a flow rate of 0.9 mL/min. The calibration curves were linear over the range of 0.125–1 µmol/L, 0.625–20 mmol/L, and 1.25–10 mmol/L for neopterin, uric acid, and creatinine levels, respectively. The inter-assay and intra-assay coefficients of variation were <5%.

### 5.4. Visual Analog Scale (VAS)

Subjective mood, general wellness/malaise, restfulness/tiredness, headaches, sleepiness, and nausea were evaluated using a 0–100 mm visual analog scale (VAS). This scoring system was previously suggested for assessing discomfort and/or general malaise [85]. Based on its usefulness in performing other clinical evaluations, VAS was considered suitable to test the subjective perception of normobaric hypoxia effects [37,86].

### 5.5. Statistical Analysis

The normality of the results was verified using the Shapiro–Wilk test. Comparisons between results at different times and the baseline were carried out using repeated-measure one-way ANOVA tests when the results had a Gaussian distribution. Otherwise, a Friedman test was applied. Comparisons between the 10% and 15% exposure groups were performed using an unpaired *t*-test (parametric) or Mann–Whitney (nonparametric) test. All data are presented as mean ± standard deviation (SD). All statistical tests were performed using a standard computer statistical package, GraphPad Prism version 9.5.1., for PC (GraphPad Software, San Diego, CA, USA). A threshold of *p* < 0.05 was considered statistically significant. The sample size required for a repeated-measure analysis of variance (Friedman) was calculated using G*power calculator 3.1.9.7 software (Heinrich-Heine-Universität, Düsseldorf, Germany) (effect size = 0.65, alpha error = 0.05, power = 0.80), and the requisite number of participants for this study was six in each group, which parallels previous studies [87].

## 6. Conclusions

A short duration (60 min) of extreme and moderate hypoxia led to a significant increase in the production of reactive oxygen species (ROS), peaking 30 min after exposure and slowly recovering after 48 h.

Renal function was astonishingly quickly activated to counteract such hypoxic stimuli, using the antioxidant action of uric acid and creatinine to counteract ROS and NOx, together with TAC significant variations.

We also found an increase in lipid peroxidation caused by ROS, starting at 2 h post-exposure.

Finally, we observed an immune system stimulation through increasing concentrations of neopterin. The rapid increase in neopterin following the ROS peak suggests a counteraction to the inflammasome complex, favoring the NRF2 pathway. Interestingly, neopterin modulation is inversely proportional to the importance of hypoxia. These small variations are very interesting as preconditioning tools.

Overall, this study provides insights into the mechanisms through which acute hypoxia affects various bodily functions and how the body responds to hypoxia-induced oxidative stress. It also highlights the protective mechanisms that humans have in place to maintain redox homeostasis in response to environmental stressors.

## Figures and Tables

**Figure 1 ijms-24-10188-f001:**
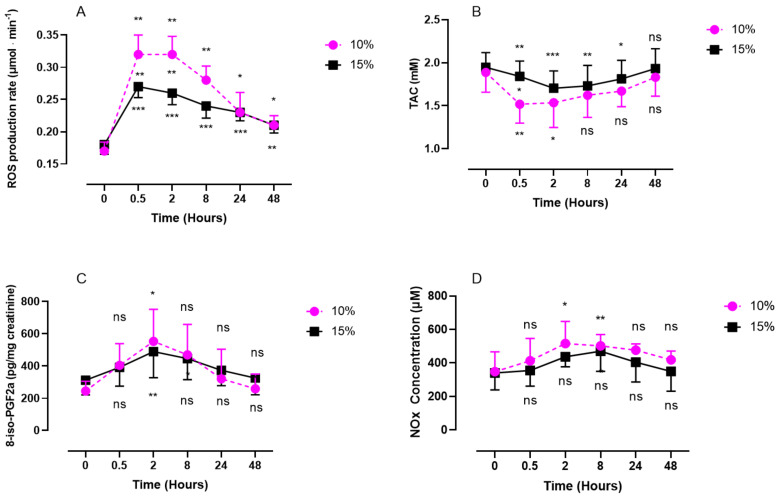
Evolution of ROS production rate (**A**), antioxidant response (TAC) (**B**), 8-iso-PGF2a (**C**), and NOx (**D**) after 60 min of mild (15%, *n* = 8) or severe hypoxia (10%, *n* = 6). Results are expressed as mean ± SD. T0 represents the pre-exposure baseline. Intra-group comparisons between results at T0 and each other time point are represented above and below the respective curves. Inter-group comparisons between 10% and 15% of oxygen exposure when significant are shown between the two curves (ns: not significant; *: *p* < 0.05, **: *p* < 0.01, ***: *p* < 0.001).

**Figure 2 ijms-24-10188-f002:**
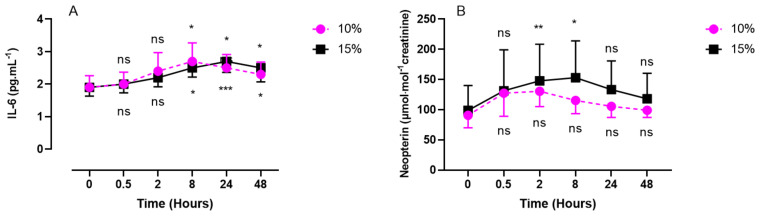
Evolution of the inflammatory response after 60 min of mild (15%, *n* = 8) or severe hypoxia (10%, *n* = 6) for interleukin-6 (IL-6) (**A**) and neopterin (**B**). Results are expressed as mean ± SD. T0 represents pre-exposure values. Intra-group comparisons between results at T0 and each other time point are represented above and below the respective curves. (ns: not significant; *: *p* < 0.05; **: *p* < 0.01; ***: *p* < 0.001).

**Figure 3 ijms-24-10188-f003:**
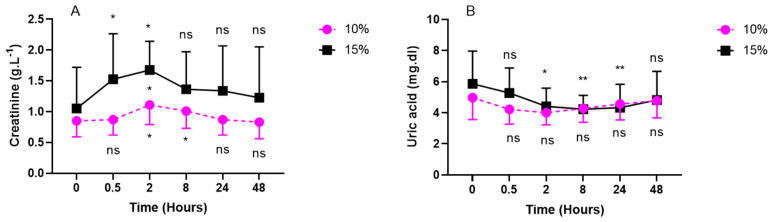
Evolution of urinary markers after 60 min of mild (15%, *n* = 8) or severe hypoxia (10%, *n* = 6) for creatinine (**A**) and uric Acid (**B**). Results are expressed as mean ± SD. T0 represents pre-exposure values. Intra-group comparisons between results at T0 and each other time point are represented above and below the respective curves. Inter-group comparisons between 10% and 15% of oxygen exposure when significant are shown between the two curves (ns: not significant; *: *p* < 0.05, **: *p* < 0.01; intragroup: Friedman; intergroup: unpaired *t*-test).

**Figure 4 ijms-24-10188-f004:**
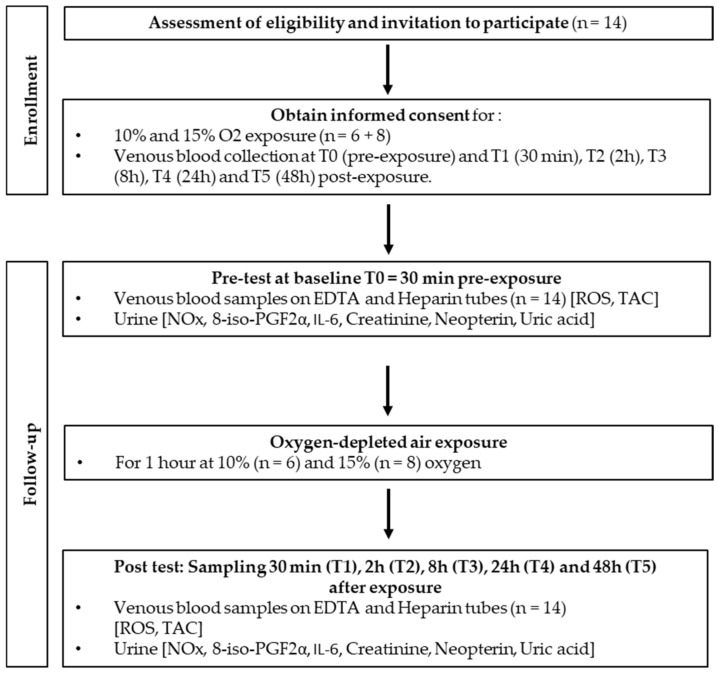
Experimental flowchart.

## Data Availability

Data are available at request from the authors.

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
