# Peer review of "Oxidative Stress Response Kinetics after 60 Minutes at Different Levels (10% or 15%) of Normobaric Hypoxia Exposure"

_ijms, 2023, doi:10.3390/ijms241210188_

Round 1

Reviewer 1 Report

The manuscript  ”Oxidative Stress Response's Kinetics after 60 Minutes at Different (10% or 15%) Normobaric Hypoxia Exposures” provides valuable insights into the effects of acute hypoxia on various bodily functions, and how the body responds to hypoxia-induced oxidative stress. However, after careful review, I recommend a major revision to address several points that could strengthen the overall quality and impact of the study. Please find below my point-by-point recommendations:

1. Participant Selection: The sample size of 14 participants is relatively small, which may limit the generalizability of the findings. Consider increasing the sample size or providing a more detailed justification for the chosen sample size.

2. Randomization: More information is needed on the randomization process. How were participants assigned to the 10% and 15% oxygen groups? Providing additional details would enhance the transparency of your study design.

3. Demographic Information: Please include more detailed demographic information about the participants, such as ethnicity, medical history, and lifestyle factors (e.g., smoking, exercise). These factors could potentially influence the study outcomes.

4. Control Group: The study lacks a control group with participants exposed to normoxic conditions. Including a control group would help establish a baseline for the physiological changes measured.

5.  Oxygen Administration: More explanation is needed on why oxygen-depleted air was administered for only 60 minutes. Justify this choice based on previous studies or pilot experiments.

6.  Blood Sample Collection: Consider explaining why specific time points were chosen for blood sample collection. The reasoning behind these choices can help readers understand the study design better.

7.  ROS Measurement: While EPR is a reliable method for ROS detection, it would be beneficial to validate the results with additional methods, such as fluorescence or chemiluminescence assays, to further strengthen the results.

8.  Urine Sample Analysis: It would be beneficial to provide a rationale for the chosen urinary biomarkers and how they relate to hypoxia and oxidative stress.

9.  VAS Measures: The use of a VAS for subjective measures is a good choice, but consider supplementing these with additional validated questionnaires to get a more comprehensive view of participants' subjective experiences.

10. Statistical Analysis: Please provide the exact p-values, not just whether they were above or below the threshold of significance. This would allow readers to better assess the strength of the findings.

11. Effect Sizes: In addition to p-values, consider reporting effect sizes for all statistical comparisons to provide a measure of the magnitude of the differences observed.

12. Confounding Factors: Address potential confounding factors that may have influenced the results, such as participants' diet, physical activity levels, and sleep patterns during the study period.

13. Long-term Effects: While the study provides valuable information about the short-term effects of hypoxia, the long-term effects remain unclear. Consider conducting follow-up assessments to examine any lasting impacts of hypoxia exposure.

14. Data Availability: Consider making the raw data available in a public repository to enhance the transparency and reproducibility of your study.

15. Interpretation of Results: The interpretation of results in the conclusion could be more detailed. For instance, the link between the rapid increase in neopterin following the ROS peak and its relation to the NRF2 pathway could be explained more explicitly.

Overall, your study has the potential to contribute significantly to our understanding of the body's response to acute hypoxia.

 Moderate editing of English language

Reviewer 2 Report

The paper “Oxidative Stress Response's Kinetics after 60 Minutes at Different (10% or 15%) Normobaric Hypoxia Exposures” addresses a timely and relevant topic. I believe that the manuscript could make a valuable contribution to our understanding of the mechanisms by which acute hypoxia affects various bodily functions and how the body sets up the protective mechanisms to maintain redox homeostasis in response to oxidative stress.  At the same time, I identified several issues that require the authors’ attention. The major points they should consider are the following:

1. Line 285-286 – “For TAC, we used the spin trap DPPH· (2,2-diphenyl-1-picrylhydrazyl), a free radical compound soluble and stable in ethanol”.

Question (1a). Which antioxidant proteins, usually localized in the body cells (mitochondria, cytosol, SOD, catalase, glutathione peroxidase/reductase, etc.) of all organs, can be determined by the DPPG method if antioxidant proteins are precipitated in an alcohol reaction medium necessary for DPPH dissolution? In this case, the DPPH method is not suitable for measuring plasma antioxidant activity (PMID: 23572765).

Question (1b). In addition, how can TAC of plasma (measured by any other method), be used to talk about the possibility of oxidative stress or its suppression in tissues, if SOD, catalase and other enzymes appear in plasma only as a result of the disruption of blood cells, mainly red blood cells? Therefore, the discussion of the results on Line 200-201- “All renal countermeasures described seem to have an efficient antioxidant role in these short hypoxic exposures since no significant TAC variations have been found like being reported in longer hypoxia”- seems inadequate. Why kidney but not other body tissues?

2. Line 154-156 “…we observe an increase in NOx production with a peak at 2 h and a higher level of significance at 8. This suggests that the activity of proliferation, sprouting, migration in the VEGF pathway is dependent on hypoxia and relative hyperoxia [42]”

Question (2a). Why does “an increase in NOx (NO2+NO3) production” lead to the suggestion that “… the activity of proliferation, sprouting, migration in the VEGF pathway is dependent on hypoxia and relative hyperoxia [42]?” A reference to article 42 mentions a coordination between the NOX (NADPH oxidase) isoforms necessary for angiogenesis that, in particular, is promoted by hypoxia NOX4-derived hydrogen peroxide, ultimately leading to the activation of NOX2 and VEGF receptors and stimulating EC proliferation. Did the authors determine the concentration of H2O2? In the text of this paper, no study’s data related to it are presented.  The observed increase in NOx (NO2+NO3) (in urine/blood samples?) therefore need to be interpreted with caution and cannot serve as a basis for association with the VEGF pathway.

3. Line181-182 «Interestingly, we observe an increase in 8- iso-PGF2α starting, which is a marker of lipid peroxidation by ROS [53, 54], at 2 hours post-exposure. Lipid peroxidation has its peak between 2 and 8 hours…»

Question (3). In what structure does lipid peroxidation enhance? Lipid peroxidation occurs under conditions where reactive oxygen species (ROS) readily attack lipids on cell membranes. Intracellular (not in plasma!) pro- and antioxidant imbalance leads to oxidative stress, resulting in lipid peroxidation.

Minor points:

1.   Figure captions are short and incomprehensible.

2.     Line 295- NOx (NO2+NO3) concentrations were determined in urine samples

Line 271-272- Figure 7. NOx (NO2+NO3) concentrations were determined in venous blood samples

Line 106-107 «Figure 1D. Evolution of NOx ...». Where was the NOx concentration measured? In urine or blood? This should be indicated in the figure caption.

3. Line 109-110 «Fig 1 Comparison between 10% and 15% oxygen exposure are shown between the 2 shapes. *: p < 0.05…» Which groups and which parameters are compared to each other? The reader should not search for this information in the text. It is necessary to expand the captions for figures.

In general, the data obtained is unconvincing, their discussion is superficial, information necessary for understanding of the “mechanisms by which acute hypoxia affects various bodily functions and how the body responds to hypoxia-induced oxidative stress” is insufficient.

The total amount of biomarkers measured in body fluids such as plasma reflects only the balance between the rate of their formation (or intake with food) and excretion and, therefore, cannot be interpreted as "oxidative stress" in tissues.

In its current form, this manuscript is not acceptable for publication. However, taking into account that the topic of the paper is interesting and important, I encourage the authors to revise and resubmit their manuscript.

Round 2

Reviewer 1 Report

The manuscript was significantly improved, so I conclude for Accept in present form.

Author Response

see uploaded file.

Reviewer 2 Report

Dear authors,

Below please find the comments in reply to your answers.

Line 285-286 The authors gave no answer to question (1a): Which antioxidant proteins, usually localized in the body cells (mitochondria, cytosol, SOD, catalase, glutathione peroxidase/reductase, etc.) of all organs, can be determined by the DPPG method if antioxidant proteins are precipitated in an alcohol reaction medium necessary for DPPH dissolution? That is why I want to draw the authors’ attention that TAC parameters of plasma are not suitable for the assessment of antioxidant status of the tissues.

The same concerns Question (1b). To avoid problems related to gender and daily changes in the SOD activity, it is necessary to have a female group and a male group (n greater than 3) and measure the SOD activity, for example, in erythrocytes but not in plasma in which enzymes appear only during cell damage.

As to Question (3). As in above described cases, analysis of lipid peroxidation in tissues using data on plasma is not informative.

According to the titles of published papers of the authors, a commonly accepted term is oxidative stress. But “The total amount of biomarkers measured in body fluids such as plasma reflects only the balance between the rate of their formation (or intake with food) and excretion and, therefore, cannot be interpreted as "oxidative stress" in tissues”.

On a whole, the minus of this work which is related to discussion of an antioxidant status of tissues using data collected from plasma measurements is still present. In this context, the mechanisms by which acute hypoxia affects various bodily functions the authors reported about and how the body sets up the protective mechanisms to maintain redox homeostasis in response to oxidative stress remain obscure.

In my view, in the current form, the article has no scientific value and I can not recommend it for publication.

Author Response

See uploaded file.
